# Identification of Targetable Lesions in Anaplastic Thyroid Cancer by Genome Profiling

**DOI:** 10.3390/cancers11030402

**Published:** 2019-03-22

**Authors:** Naveen Ravi, Minjun Yang, Sigurdur Gretarsson, Caroline Jansson, Nektaria Mylona, Saskia R. Sydow, Eleanor L. Woodward, Lars Ekblad, Johan Wennerberg, Kajsa Paulsson

**Affiliations:** 1Department of Laboratory Medicine, Division of Clinical Genetics, Lund University, SE-221 84 Lund, Sweden; naveen.ravi@med.lu.se (N.R.); Minjun.yang@med.lu.se (M.Y.); Caroline.jansson@med.lu.se (C.J.); Saskia.sydow@med.lu.se (S.R.S.); Eleanor.woodward@med.lu.se (E.L.W.); 2Division of Otorhinolaryngology/Head and Neck Surgery, Clinical Sciences, Lund University and Skåne University Hospital, SE-221 85 Lund, Sweden; stg72@live.com (S.G.); Johan.wennerberg@med.lu.se (J.W.); 3Division of Oncology and Pathology, Clinical Sciences, Lund University and Skåne University Hospital, SE-221 85 Lund, Sweden; Nektaria.mylona@skane.se (N.M.); Lars.Ekblad@med.lu.se (L.E.)

**Keywords:** anaplastic thyroid cancer, whole exome sequencing, RNA-sequencing, formalin-fixed paraffin embedded tissues, fusion genes, somatic mutations, copy number alterations, *CCNE1*

## Abstract

Anaplastic thyroid cancer (ATC) is a rare and extremely malignant tumor with no available cure. The genetic landscape of this malignancy has not yet been fully explored. In this study, we performed whole exome sequencing and the RNA-sequencing of fourteen cases of ATC to delineate copy number changes, fusion gene events, and somatic mutations. A high frequency of genomic amplifications was seen, including 29% of cases having amplification of *CCNE1* and 9% of *CDK6*; these events may be targetable by cyclin dependent kinase (CDK) inhibition. Furthermore, 9% harbored amplification of *TWIST1*, which is also a potentially targetable lesion. A total of 21 fusion genes in five cases were seen, none of which were recurrent. Frequent mutations included *TP53* (55%), the *TERT* promoter (36%), and *ATM* (27%). Analyses of mutational signatures showed an involvement of processes that are associated with normal aging, defective DNA mismatch repair, activation induced cytidine deaminase (AID)/apolipoprotein B editing complex (APOBEC) activity, failure of DNA double-strand break repair, and tobacco exposure. Taken together, our results shed new light on the tumorigenesis of ATC and show that a relatively large proportion (36%) of ATCs harbor genetic events that make them candidates for novel therapeutic approaches. When considering that ATC today has a mortality rate of close to 100%, this is highly relevant from a clinical perspective.

## 1. Introduction

Anaplastic thyroid cancer (ATC) is an extremely aggressive tumor, with close to 100% mortality and no available cure [1,2,3,4]. The lack of curative treatments for ATC makes it important to understand the underlying tumorigenesis of this disease; however, the genomic landscape of ATC has not yet been fully delineated.

Cytogenetic analysis and array comparative genome hybridization (aCGH) of ATC has revealed high levels of aneuploidy, with chromosome numbers ranging from 65–120; however, these studies were performed on small cohorts of samples [5,6,7,8,9,10,11]. Recently, Pozdeyev et al. found amplifications involving *KIT* in 4q12 (4% of cases), *CCNE1* in 19q12 (4% of cases) and *CD274* (previously *PD-L1*), *PDCD1LG2* (previously *PD-L2*), and *JAK2* in 9p24 (3% of cases) using targeted sequencing. As regards to structural rearrangements, a *STRN*-*ALK* translocation has been found in one case of ATC [12], but no recurrent gene fusions have been identified. Genes that are recurrently mutated in ATC include *TP53* (25–60% of cases), *BRAF* (25–90%), *USH2A* (20%), *NRAS* (15–20%), *PTEN* (15%), *NF1* (10–35%), *PIK3CA* (10–20%), *EIF1AX* (10%), *ATM* (8%), *HRAS* (7%), *KRAS* (5–10%), and *CTNNB1* (5%) [13,14,15,16,17,18,19,20,21,22]. Furthermore, *TERT* promoter mutations, leading to *TERT* expression, are seen in 15–75% of cases [13,17,18,21,22,23].

Most next generation sequencing studies of ATC have been done by targeted sequencing of custom gene panels [13,14,15,16,17,18,19]. In total, only 41 primary ATCs that were investigated with whole exome seuencing (WES) have been previously published [20,21,24], and no RNA-sequencing (RNA-seq) that is aimed at fusion gene detection is available in the literature. As the genomic landscape of ATC has thus not yet been fully explored, we applied WES and RNA-seq on primary tumor samples to identify the novel genetic events that contribute to ATC tumorigenesis and that may be used as therapeutic targets.

## 2. Results

### 2.1. Genomic Amplifications Are Common in ATC

Copy number analysis could be undertaken based on WES data in 10 cases (Table 1). Seven of the analyzed cases had large variations in chromosome copy number, as well as variant allele frequencies (VAFs) that were suggestive of polyploidy, whereas three appeared to have near-diploid genomes (Figure 1a, Appendix A). A median of 16 breakpoints, defined as a change in copy number state, were detected per case (range 5–43), with a high proportion (31/187; 17%) of breakpoints occurring in centromeres; all of the cases had at least one breakpoint in a centromere (range 1–7; Figure 1b, Appendix A). Chromosome 8 displayed a pattern of loss of 8p and concurrent gain of 8q, with breakpoints in the centromere in six cases (60%), possibly indicating isochromosome 8q. Two additional cases had gain of the whole chromosome 8, making gain of 8q present in 8/10 (80%) investigated cases (Appendix A).

High-level amplifications, defined as a gain of more than three extra copies over the baseline level, were seen in eight of ten (80%) cases, with recurrent amplifications in 19q12 and 19q13 (Appendix A). The 19q12 amplification was seen in three cases (30%) with the minimally gained region chr19:30020714-30649335, including the *POP4*, *PLEKHF1*, *C19orf12*, *CCNE1*, and *URI1* genes. All of these genes were highly expressed in the cases with amplification as compared to cases without amplification (Figure 1c). In addition, one of the cases where no copy number analysis had been performed displayed high expression for all of these genes. Furthermore, two cases (20%) had 19q13 amplifications, with the minimally gained region chr19:36494128-36585117, including the *ALKBH6*, *CLIP3*, and *THAP8* genes. Of these, only *CLIP3* showed increased expression in cases with the amplification (Figure 1c). Among the amplifications that were only seen in single cases, case 2 had an amplification in 7q21 with the minimally gained region chr7:91974285-92987750, including the *GATAD1*, *PEX1*, *RBM48*, *FAM133B*, *CDK6*, *SAMD9*, *SAMD9L*, and *HEPACAM2* genes. All of these genes, except for *SAMD9*, *SAMD9L* and *HEPACAM2*, were highly expressed in this case (Figure 1c). Furthermore, case 2 had an amplification of chr7:18330180-19461461, including *TWIST1*, which was highly expressed in this case (Figure 1c).

### 2.2. Multiple Non-Recurrent Fusion Genes in ATC

A total of 21 fusion genes were identified in 5/12 cases that were investigated with RNA-seq (Table 2). Of these, 15 were seen in case 5, whereas cases 3 and 7 had two fusion genes each and cases 12 and 14 had one fusion gene each. Nine of the fusion genes were in-frame and twelve out-of-frame. None of the fusion genes were recurrent, but *FN1* was involved in two different out-of-frame fusions. For *MLXIP*/*PTEN* and *EP400*/*NCOR2* fusion genes, the reciprocal *PTEN*/*MLXIP* and *NCOR2*/*EP400* fusions, respectively, were seen; no other reciprocal fusions were detected.

### 2.3. Somatic Mutations in ATC

WES detected a total of 7478 somatic coding mutations in the eight cases with matched normal samples for analysis, with a median of 60 mutations per case (range 28–6863) (Appendix A). Excluding case 12, which had 6863 mutations, the remaining ten cases had a median of 52 mutations per case (range 28–247). For the three cases with no matched normal, a total of 245 mutations remained after filtering, with a median of 87 mutations per case (range 58–99) (Appendix A). The most commonly mutated gene was *TP53* (6/11 cases; 55%), followed by mutations in the *TERT* promoter (four cases; 36%), *ATM* (three cases; 27%), and *ARID2*, *BRAF*, *FANCA*, *INPP4B*, *MAP3K1*, *NF2*, *PIK3CA*, *RB1*, *SMARCA4*, and *TET2* (two cases each; 18%) (Figure 2, Appendix A). Furthermore, single cases (9%) had mutations in *NRAS* and *HRAS* (Figure 2, Appendix A). In addition, case 12 displayed a mutation in *CCNE1*, which was classified as “tolerated” by SIFT and “benign” by PolyPhen2; it is thus not clear whether this had a pathogenetic effect.

### 2.4. Associations Between Genetic Aberrations and Clinical Parameters

To see whether the genetic aberrations were associated with any clinical parameters, we performed Fisher’s exact two-sided test for the recurrent abnormalites that are listed in Figure 2 vs. nodal stage and presence of distant metastases. No significant associations were detected.

### 2.5. Mutational Signatures in ATC

The analysis of mutational signatures could be performed in all 11 cases that were subjected to WES. The most common mutational signature was 1A/B, associated with aging, seen in six cases (55%; Appendix A). Five cases (45%) displayed signature 6, associated with defective DNA mismatch repair, four (36%) displayed signatures 2 or 13, associated with activity of the AID/APOBEC family of cytidine deaminases, four (36%) displayed signature 3, associated with failure of DNA double-strand break-repair by homologous recombination, four (36%) displayed signature 4, associated with tobacco exposure, three (27%) displayed signature 7, associated with ultraviolet light exposure, and two (18%) displayed signature 11, associated with exposure to alkylating agents (Appendix A, https://cancer.sanger.ac.uk/cosmic/signatures). Furthermore, case 12 displayed signature 14, which is associated with very high numbers of somatic mutations. Additionally, signature 17, which was of unknown etiology, was seen in five cases (45%).

### 2.6. Microsatellite Instability Is Rare in ATC

WES data was analyzed to check whether ATC exhibited microsatellite instability (MSI). Only one (9%) of the eleven cases had a value that was close to the cut-off for MSI (Figure 3). Notably, this case (#12) also had mutations in *MLH1* and *MSH2*.

## 3. Discussion

In spite of ATC being one of the most fatal malignancies, its genetic background has not yet been fully explored. Here, we present data on copy number, fusion genes, and mutations from fourteen primary ATC samples, obtained before chemo- or radiotherapy.

Copy number analysis was done based on WES, providing a variable resolution across the genome depending on gene density. In line with previous studies using chromosome banding and aCGH [5,6,7,8,9,10,11], we found that at least 7/10 (70%) tumors were polyploid based on variable copy number and VAF. Furthermore, a high frequency of breakpoints in the centromeric regions were seen, similar to what we have previously reported in the ATC cell lines [26]. This has been proposed to be a sign of chromosomal instability that can arise either due to the formation of breakage-fusion-bridge cycles that are caused by telomere dysfunction or due to mitotic spindle defects [27,28]. On the chromosomal level, most of the chromosomes displayed variable copy numbers between cases. The exception was chromosome 8, which showed a loss of 8p relative to the baseline copy number in 6/10 (60%) cases and gain of 8q in 8/10 cases (80%). Thus, it is likely that a gain of 8q is a driver event in ATC; however, since this is a large region containing >1500 genes, it was not possible to identify the target gene(s).

A relatively high number of amplifications was seen. Three cases (#2, #3, and #13) harbored amplification of 19q12 with *CCNE1* as the putative target gene. CCNE1 promotes progression into the S phase of the cell cycle by interacting with cyclin dependent kinases (CDKs) [29]. *CCNE1* amplifications were recently reported in ATC [17] and they are known to occur in multiple malignancies, including ovarian, breast, and gastric cancer [30,31,32,33]. Apart from being highly expressed in all cases with the amplification, one additional case (#5) without copy number data displayed high expression of *CCNE1* and also likely had amplification. Furthermore, case 12 displayed a somatic mutation in *CCNE1*. However, this was classified as non-pathogenic and possibly a passenger event. Thus, in total, 4/14 (29%) of our cases displayed an amplification of *CCNE1*; a frequency much higher than the 4% of cases that were reported by Pozdeyev et al. [17], although this difference could be due to the small size of our cohort. Interestingly, one of our cases (#2) also had the amplification and overexpression of *CDK6*, active in the same cellular processes as CCNE1 [34]. Although there are no drugs that are available that specifically targets CCNE1, multiple inhibitors for CDKs are under development and could be viable treatment options in *CCNE1*- and *CDK6*-amplified ATC [29,35].

Amplification of *TWIST1* was seen in case 2. TWIST1 is a transcription factor that is involved in the epithelial-to-mesenchymal transition (EMT) pathway and it is frequently overexpressed in cancer [36]. Preclinical studies in mouse models of *KRAS*-mutant lung cancer suggest that harmine, a β-carboline alkaloid, can be used to target TWIST1 with high efficacy [37].

A total of 21 fusion genes were identified, none of which were recurrent. There was a large variation in the number of fusion genes detected per case, with case 5 displaying 16 fusion genes and the remaining eleven cases investigated with RNA-seq having 0–2 fusion genes (Table 2). This is similar to what we have previously reported in ATC cell lines [26], and suggest that a large proportion of the fusion genes detected may be passenger events resulting from chromosomal breaks. Notably, *FN1* was involved in two different fusion genes—a *FN1/PABPC1* in case 5 and a *USP46/FN1* in case 7. *FN1* is a component of the extracellular matrix and it is recurrently involved in in-frame fusion genes as the 5′ partner in *FN1-ALK* and *FN1-FGFR1* in soft tissue tumors [38,39]. However, the *FN1* fusions that were detected here were both out-of-frame and therefore likely to result in a loss of the normal function of *FN1*. Furthermore, one case harbored an in-frame *PTEN*/*MLXIP* fusion, which has not been previously reported. *PTEN* is a well-known tumor suppressor gene that regulates the PI3K/AKT pathway [40]. Both in-frame and out-of-frame *PTEN* fusion genes have been reported to occur at a low frequency in various malignancies; the pathogenetic outcome is generally considered to be the loss of normal *PTEN* function [41,42,43].

The pattern of somatic mutations found in our study was similar to what has been previously reported [13,14,15,16,17,18,19,20,21,22,23]. An analysis of mutational signatures showed a high incidence (>30% of cases) of processes that are involved with normal aging, defective DNA mismatch repair, AID/APOBEC activity, failure of DNA double-strand break repair, and tobacco exposure. This agrees well with the results from Pozdeyev et al. [17], who reported that defective DNA mismatch repair and the activation of AID/APOBEC was common in ATC, based on targeted sequencing data from 24 cases, and also with the data from Dong et al. [24] from WES in five cases. Taken together, though, the mutational processes occurring in ATC appear to differ between cases (Appendix A), suggesting different underlying etiologies.

Case 12 was an outlier in terms of having a much higher number of somatic mutations (6863 vs. a median of 52 for the remaining cases), as well as occurring in a relatively young (49 years) patient. This case also showed a borderline value for microsatellite instability and it had mutations in both *MLH1* and *MSH2*, involved in DNA mismatch repair (Figure 2 and Figure 3). Notably, case 12 also exhibited mutational signature 14, which was recently shown to be caused by a combination of loss of polymerase proofreading due to mutations in *POLE* or *POLD1* and defective mismatch repair [44]; in line with this, case 12 had two mutations in *POLE* (Appendix A). Microsatellite instability has been proposed to be a marker for a favorable response to PD1 blockade therapy [45]; thus, this treatment could have been a viable therapeutic option for this patient.

Pozdeyev et al. [17] recently suggested that ATC may be divided into three different subtypes that are based on the mutational pattern: (1) tumors with *BRAF* V600E mutations, together with *PIK3CA*, *AKT1* or *ARID2* mutations, (2) tumors with *NRAS* mutations and *CCNE1* amplification, and (3) tumors with a high mutational burden and *MSH2*/*MLH1* mutations. In our cohort, cases 9 and 11 could be classified as type 1 according to this system based on having concurrent *BRAF* V600E and *ARID2*/*PIK3CA* mutations. Cases 2, 3, 5, and 13 had *CCNE1* amplification corresponding to type 2, none of which had *NRAS* mutation, whereas case 1 had an *NRAS* mutation; all of these could tentatively be classified as type 2. Finally, case 12 had a very high number of somatic mutations (*n* = 6863) and mutations in both *MSH2*/*MLH1,* agreeing well with type 3. The remaining cases could not be classified as either of these types based on the mutations.

Taken together, we show that a relatively large proportion (5/14 cases; 36%) of ATC harbor genetic events that make them suitable for novel therapeutic approaches, including CDK and TWIST1 inhibition, as well as PD1 blockade therapy. When considering the dismal prognosis of this disease, this should be addressed in future clinical trials.

## 4. Material and Methods

### 4.1. Patient Samples

The study initially included a total of 23 cases of ATC, which were selected on the basis of not having obtained chemo- or radiotherapy treatment prior to sampling, as well as on sample availability. Twenty-two formalin-fixed, paraffin-embedded (FFPE) samples with hematoxylin and eosin stained tumor sections were obtained from the Pathology Department, Laboratory Medicine, Skåne, Sweden. Furthermore, a fine-needle aspirate that was obtained at ATC diagnosis and a paired peripheral blood sample was included from one additional patient (case 1). A pathologist reviewed all of the FFPE blocks to confirm the presence of ATC and to determine tumor cell content. Cases with <30% tumor cells based on the pathologist’s estimate (*n* = 5), and cases with no copy number aberrations or mutations that were detected by WES (*n* = 4) were excluded from further analyses, leaving 14 cases (Table 1). Three 10 μm sections were cut from each tumor and parts containing tumor and adjacent normal tissue were manually microdissected when possible. Sections were immediately put in deparaffinization solution (Qiagen, Valencia, CA, USA), followed by DNA and RNA extraction with the all prep DNA/RNA FFPE Kit (Qiagen), according to the manufacturer’s recommendations. For the fine needle aspirate and the matching peripheral blood sample, DNA and RNA extractions were performed with the all prep DNA/RNA kit (Qiagen), according to the manufacturer’s recommendations. The Ethical Review Board of Lund University approved the study (No. 2016/51, 1 February 2016).

### 4.2. Whole Exome Sequencing

WES was performed on eight matched tumor-normal samples and three tumor samples without matching normal samples (Table 1). DNA damage that was caused by formaldehyde fixation was repaired with the PreCR Repair Mix (New England Biolabs, Ipswich, MA, USA) according to the manufacturer’s recommendations. Genomic DNA was sheared via sonication while using an S220 focused-ultrasonicator (Covaris, Woburn, MA, USA) and DNA libraries were constructed using the TruSeq Exome Kit (Illumina, San Diego, CA, USA), according to the manufacturer’s recommendations. The libraries were sequenced using the High Output Kit (150 cycles) on a NextSeq500 (Illumina). Raw reads were aligned to the human reference genome hg19 with the BWA-MEM algorithm [46]. Picard (http://broadinstitute.github.io/picard) was used to remove the PCR duplicates and local realignment around the indel region was performed using GATK [47]. Copy number aberrations were identified by cnvkit [48] and manually annotated; the resulting data from case 12 were too noisy for interpretation, and this case was hence excluded from copy number analysis. For cases with matched normal samples, the somatic mutations were identified using MuTect2 [49] with default settings. For tumor samples without matched normal samples, variations were identified by GATK Unified Genotyper and annotation parameters QD (variant confidence/quality by depth) < 2.0, MQ (root mean square mapping quality) < 40.0, FS (Fisher strand) 60.0, HaploTypeScore > 13.0, MQRankSum < −12.5, and ReadPosRankSum < −8.0 were used to filter the low quality variations [50]. High quality variants were further filtered by 1000 Genomes (20110521 release), ESP6500, ExAC, CG46 (popfreq_max_20150413), and 170 million variants (kaviar_20150923) provided by ANNOVAR [51] to remove the potential SNP sites. Annotation of variants were carried out with ANNOVAR [51]. The lists of somatic mutations were further filtered for a minimum coverage of 15 reads, keeping only non-synonymous mutations that were supported by ≥5 reads and a mutant allele frequency of ≥5%. Mutation signatures were identified with DeconstructSigs [52]. The genomic landscape plot was generated using GenVisR [53]. MSIsensor [54] was used for analyzing microsatellite instability. Only heteropolymer sites were included in this analysis.

### 4.3. RNA Sequencing

RNA-seq was performed on 12 primary ATC cases (Table 1) and normal tissue from four thyroids. For the tumor cases, the quantity and purity of RNA was measured with NanoDrop (Thermo Fisher Scientific, Waltham, MA, USA) and the quality on a 2100 Bioanalyzer (Agilent Technologies, Palo Alto, CA, USA) to check for the fraction of RNA fragments that were greater than 200 nt (Dv200). mRNA libraries were constructed with an input of 20–50 ng of RNA, depending on the Dv200 value using the TruSeq RNA Access Library Prep Kit (Illumina), according to the manufacturer’s recommendations. Constructed libraries were sequenced using Illumina’s High Output Kit (150 cycles) on an Illumina NextSeq500. Identifications of fusion transcripts were performed using FusionCatcher [55] and InFusion [56] from raw fastq files. The list of fusion genes was filtered to remove chimeras that were identified as read-through transcripts, pseudogenes, unannotated genes, and fusions between gene family members, as well as by keeping only fusions that had unique spanning reads ≥ 3 (FusionCatcher) and ≥ 20 (InFusion). For expression analysis, RNA sequencing data were processed using the TCGA mRNA-seq pipeline (https://docs.gdc.cancer.gov/Data/Bioinformatics_Pipelines/Expression_mRNA_Pipeline/#mrna-analysis-pipeline). Briefly, the sequencing reads were aligned to the human GRCh38 genome assembly using STAR [57] and the read counts for each gene were obtained using HTSeq-count [58] and they were normalized using the fragments per kilobase of exon model per million mapped reads (FPKM) method. For fusion gene validation, RT-PCR was performed in case 14, which was the only case where cDNA could be obtained. Briefly, cDNA was synthesized while using High–Capacity cDNA Reverse Transcription Kit (Thermo Fisher Scientific) and SuperScript IV First-Strand Synthesis System (Thermo Fisher Scientific). Primers (available on request) were designed using Primer 3 (http://primer3.ut.ee/) specifically for the fusion transcript. PCR was performed according to standard methods and Eurofins Genomics sequenced the amplified products (Ebersberg, Germany).

### 4.4. Analysis of TERT Promoter Mutations

*TERT* promoter mutations were investigated according to Liu et al. [59] using the AmpliTaq Gold 360 Master Mix (Thermo Fisher Scientific).

### 4.5. Statistical Analysis

Fisher’s two-sided exact test was used to investigate whether any of the detected genetic aberrations were associated with clinical parameters. Since all cases were T4, stage IV, and had high Ki67, this analysis could only be done for nodal stage and the presence of distant metastases.

## 5. Conclusions

In conclusion, we have performed a full-scale genomic analysis of primary ATC, showing complex copy number aberrations and polyploidy, multiple fusion genes, and a high level of mutations. A high proportion of the investigated cases were found to be candidates for novel treatments, showing that genomic analyses are highly clinically valuable in ATC.

## Figures and Tables

**Figure 1 cancers-11-00402-f001:**
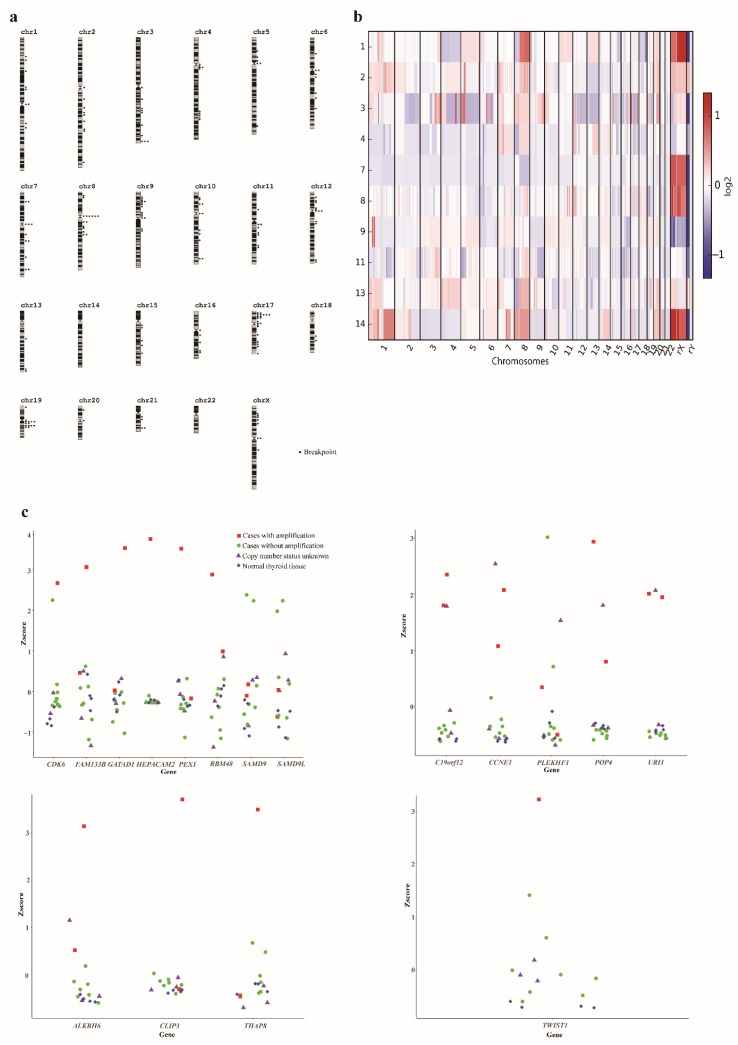
Detection of copy number variants in anaplastic thyroid cancer (ATC). (**a**) Breakpoint map of 10 primary ATC cases based on whole exome sequencing. Breakpoints were defined as a change in copy number state. A large number of breakpoints in centromeric regions were seen. (**b**) Heat map of copy number aberrations in 10 primary ATC cases. Polyploidy and large variations in chromosomal copy number were seen. (**c**) Expression of genes in amplified regions in all ATC cases and normal thyroid tissue.

**Figure 2 cancers-11-00402-f002:**
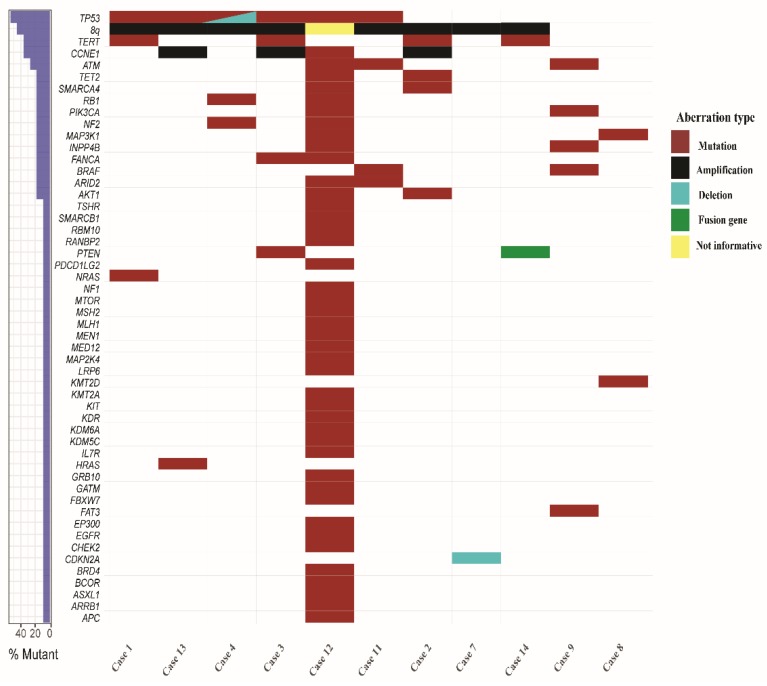
Genomic landscape of 11 cases of anaplastic thyroid cancer investigated by whole exome sequencing.

**Figure 3 cancers-11-00402-f003:**
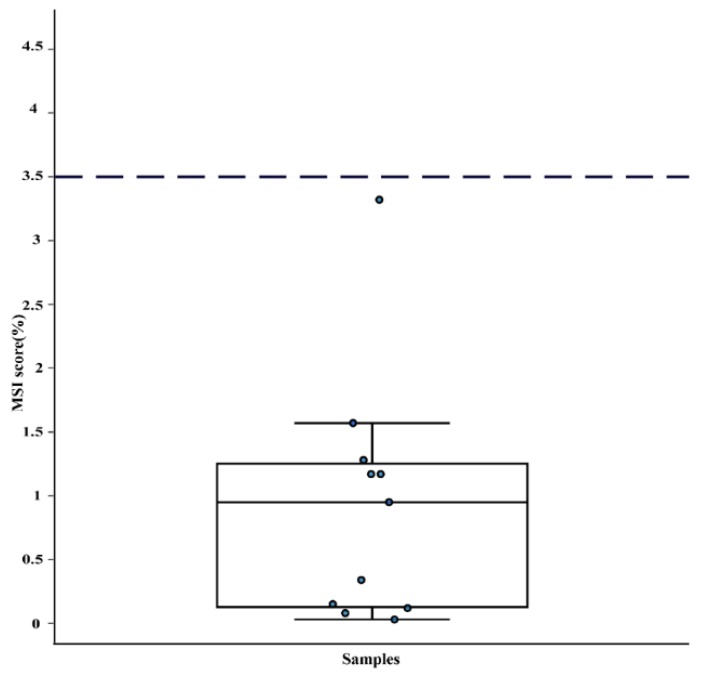
Microsatellite instability (MSI) scores for 11 anaplastic thyroid cancer cases. The dotted line represents the threshold to classify a case as MSI positive. Only one case was borderline MSI positive.

**Table 1 cancers-11-00402-t001:** Clinical data and genetic analyses of 14 cases of primary anaplastic thyroid cancer.

Case No.	Gender	Age	Tumor Size (cm)	T	N	M	Stage	Ki67 (%)	Copy Number Analysis	Fusion Gene Analysis	Mutation Analysis (Matched Normal)
1	F	71	5.5 × 4 × 7	T4b	N1	M1	IV C	N/A	Yes	Yes	Yes (yes)
2	M	70	6 × 8	T4a	N0	M0	IV A	35	Yes	Yes	Yes (yes)
3	F	73	8 × 6 × 5	T4b	N1b	M0	IV B	75	Yes	Yes	Yes (no)
4	M	64	4.6 × 4.3 × 7.4	T4b	N0	M0	IV B	90	Yes	No	Yes (yes)
5	M	64	8 × 7	T4b	N0	M1	IV B	60	No	Yes	No
6	F	72	5 × 5 × 7	T4b	N1b	M0	IV B	N/A	No	Yes	No
7	F	74	6 × 10 × 7	T4b	N1	M1	IV C	N/A	Yes	Yes	Yes (yes)
8	F	84	11.9 × 8.3 × 11.7	pT4b	pN1b	pM1	IV C	50	Yes	Yes	Yes (yes)
9	F	86	7 × 5.5 × 4.5	pT4b	No	M1	IV C	N/A	Yes	Yes	Yes (no)
10	F	70	5 × 3.5 × 5	T4b	N0	M0	IV B	50	No	Yes	No
11	M	84	8.5 × 6.5 × 5.5	T4b	N0	M1	IV C	N/A	Yes	Yes	Yes (yes)
12	M	49	7 × 7 × 5	T4b	N0	M0	IV B	N/A	No	Yes	Yes (yes)
13	M	76	4.8 × 3.7 × 8.3	T4b	N1b	M1	IV C	N/A	Yes	No	Yes (no)
14	F	63	8 × 5.5	T4b	N0	M0	IV B	30	Yes	Yes	Yes (yes)

N/A, data not available; T: size/extent of primary tumor; N, degree of spreading to regional lymph nodes; M, presence or absence of distant metastasis. TNM staging according to Sobin et al [25].

**Table 2 cancers-11-00402-t002:** Fusion genes detected in twelve cases of primary anaplastic thyroid cancer.

Case	Fusion Gene	Chromosome	Inframe/Frame-Shift	Software Identifying Fusion	Validated
3	*BGN*/*THOC7*	Xq28/3p14.1	Inframe	FusionCatcher	Not done
3	*POSTN*/*EIF3A*	13q13.3/10q26.11	Frame-shift	InFusion	Not done
5	*EP400*/*NCOR2* *NCOR2*/*EP400*	12q24.33/12q24.31	Inframe	FusionCatcher	Not done
5	*FN1*/*PABPC1*	2q35/8q22.3	Frame-shift	FusionCatcher	Not done
5	*IVNS1ABP*/*KYNU*	1q25.3/2q22.2	Inframe	FusionCatcher	Not done
5	*MYH9*/*EIF2AK3*	22q12.3/2p11.2	Frame-shift	FusionCatcher	Not done
5	*PRPF6*/*TENM3*	20q13.33/4q35.1	Inframe	FusionCatcher	Not done
5	*RAB23*/*DST*	6p11.2/6p11.2	Inframe	FusionCatcher	Not done
5	*MYH3*/*FZD4*	17p13.1/11q14.2	Inframe	InFusion	Not done
5	*TAOK1*/*NME6*	17q11.2/3p21.31	Inframe	InFusion	Not done
5	*CNTN1*/*CCZ1B*	12q12/7p22.1	Frame-shift	InFusion	Not done
5	*HELZ*/*MYH10*	17q24.2/17p13.1	Inframe	InFusion	Not done
5	*VSIG4*/*TRA2B*	Xq12/3q27.2	Frame-shift	InFusion	Not done
5	*OPHN1*/*PTRF*	Xq12/17q21.2	Frame-shift	InFusion	Not done
5	*SDC2*/*SRRT*	8q22.1/7q22.1	Frame-shift	InFusion	Not done
5	*HTRA1*/*AMZ2*	10q26.13/17q24.2	Frame-shift	InFusion	Not done
5	*GPR107*/*MYH10*	9q34.11/17p13.1	Inframe	InFusion	Not done
7	*MXI1*/*STMN1*	10q25.2/1p36.11	Frame-shift	FusionCatcher	Not done
7	*USP46*/*FN1*	4q12/2q35	Frame-shift	FusionCatcher	Not done
12	*ENO2*/*PIEZO2*	12p13.31/18p11.21	Frame-shift	InFusion	Not done
14	*MLXIP*/*PTEN* *PTEN*/*MLXIP*	12q24.31/10q23.31	Inframe	FusionCatcher	Yes

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
