# Peer review of "Identification of Targetable Lesions in Anaplastic Thyroid Cancer by Genome Profiling"

_cancers, 2019, doi:10.3390/cancers11030402_

Round 1

Reviewer 1 Report

This is an interesting original research that investigated the genetic landscape of a selected series of fourteen cases of anaplastic thyroid cancer. In the submitted study the Authors performed whole exome sequencing and RNA-sequencing to delineate copy number changes, fusion gene events and somatic mutations. The Authors found a high frequency of genomic amplifications and frequent mutations of several genes such as TP53, the TERT promoter and ATM suggesting these genetic changes in anaplastic thyroid cancer tumorigenesis. Furthermore the Authors suggest that several genomic changes may be targetable by novel therapeutic approaches (e.g. CDK inhibition).

Comments: the Introduction section is well written, figures and table are nice and representative. Discussion and results are clear and the references are up-dated.

Major criticism:                                                                                              I would suggest to organize the Material and Method section including the main clinicopathological features such as: tumor size, capsule infiltration lymph node metastases and stage of disease for each analyzed patient. These clinicopathological characteristic should be correlated with the main genomic changes found and then showed in a table in the Results section.

Author Response

This is an interesting original research that investigated the genetic landscape of a selected series of fourteen cases of anaplastic thyroid cancer. In the submitted study the Authors performed whole exome sequencing and RNA-sequencing to delineate copy number changes, fusion gene events and somatic mutations. The Authors found a high frequency of genomic amplifications and frequent mutations of several genes such as TP53, the TERT promoter and ATM suggesting these genetic changes in anaplastic thyroid cancer tumorigenesis. Furthermore the Authors suggest that several genomic changes may be targetable by novel therapeutic approaches (e.g. CDK inhibition).

Comments: the Introduction section is well written, figures and table are nice and representative. Discussion and results are clear and the references are up-dated.

Major criticism:                                                                                              I would suggest to organize the Material and Method section including the main clinicopathological features such as: tumor size, capsule infiltration lymph node metastases and stage of disease for each analyzed patient. These clinicopathological characteristic should be correlated with the main genomic changes found and then showed in a table in the Results section.

We agree that the issue of associations between genetic aberrations and clinical parameters is important and we have added the requested clinical data to Table 1. Unfortunately, since all tumors were T4, stage IV and had high Ki67, this analysis could only be done for nodal stage and presence of distant metastases. No significant associations were detected. We have added this to the text (new Results paragraph 2.4 and Materials and Methods paragraph 4.5).

Reviewer 2 Report

In this report, Ravi and colleagues analyze 14 cases of anaplastic thyroid carcinoma (ATC) by whole  exome sequencing (WES) and RNA-seq, to better define the genomic features of this tumor type. The study is relevant, since ATC is a deadly cancer and no curative treatments are available. Moreover, at variance from the well differentiated forms, such as papillary thyroid carcinoma  (PTC), that features very low mutational burden, ATC displays a more complex genomic lanscape. Previous studies identified various genetic aberrations in ATC. However, due to the extremely low frequency of ATC, studies on novel ATC cases are very important and may add relevant knowledge to the comprehension of the molecular basis of this disease.

In synthesis, the authors confirmed previous observations, regarding the frequencies of known oncogenes/tumor suppressor genes, including somatic mutations in p53, TERT promoter, BRAF, H- and N-RAS, ATM, but also found novel genes possibly involved in ATC. These include amplifications in the CCNE1 and CDK6 genes, as well as Twist amplification, and chromosome 8 gain.

This is a descriptive study, which opens new possible therapeutic opportunities for ATC. In particular, the finding that some cases show cyclin E amplification is of great interest, as cyclin E is the partner of CDK2 kinase and it has been implicated in many cancer types. CycE deregulation has been linked to DNA replication stress, and CycE, but not other cyclins, has been shown to promote chromosome instability (gains or losses), in special way together with p53 mutations. It would be interesting to further analyze ATC cases that feature both CycE amplification and p53 loss of function (case 13 and 3): in particular, the authors could assess the expression levels of CycE and p53 by IHC, as well as the levels of  mRNAs for targets of  the E2F transcription factor, that is activated by CycE/CDK2 complexes, by RT-PCR. Moreover, the authors could evaluate, by IHC, mitotic index and centrosomal aberrations by gamma-tubulin staining.

Author Response

Please see attached pdf file.

Reviewer 3 Report

The manuscript was aimed to explore the genetic pattern of anaplastic thyroic cancer, by Whole Exame Sequencing and RNA Sequencing. The article is well written, with results and discussion properly and extesively described. The results provide a new scenario for further challenges in clinical therapy of ATDs,

I point out only a minor revision. Among the supplementary files I could not find the supplementary tables (Supplementary Tables 1,2, 3). I suggest the authors to reconsider to insert these tables within the principal text, representing very important findings.

Author Response

We sincerely apologize for this error during the submission process; of course these should have been included in the submission (and we erroneously thought they were). Since these tables are so long we do not think that they can be included in the principal text, but all relevant data are summarized there.

Round 2

Reviewer 1 Report

The manuscript is now suitable for publication

Reviewer 2 Report

I am satisfied with the answer given by the authors and I believe the manuscript can be published in its present form